# Influence of the Thickness of a Nanolayer Composite Coating on Values of Residual Stress and the Nature of Coating Wear

**Alexey Vereschaka [1,\*], Marina Volosova [1], Anatoli Chigarev [2], Nikolay Sitnikov [3], Artem Ashmarin [4], Catherine Sotova [1] , Jury Bublikov [5] and Dmitry Lytkin [6]**

1   VTO Department, Moscow State Technological University STANKIN, Moscow 127994, Russia; m.volosova@stankin.ru (M.V.); e.sotova@stankin.ru (C.S.)
2   Department of Theoretical Mechanics, Belarusian National Technical University, Minsk 220013, Belarus; chigarev@rambler.ru
3   Department of Solid State Physics and Nanosystems, National Research Nuclear University MEPhI, Moscow 115409, Russia; sitnikov_nikolay@mail.ru
4   Baikov Institute of Metallurgy and Materials Science RAS, Moscow 119334, Russia; ashmarin_artem@list.ru
5   Institute of Design and Technology Informatics (IKTI) RAN, Moscow 127055, Russia; yubu@rambler.ru
6   Central Research Institute of Machine Building Technology (PJSC SPA "CNIITMASH"), Moscow 115088, Russia; ldndima@yandex.ru
*   Correspondence: dr.a.veres@yandex.ru

**Abstract:** The article discusses the influence of the thickness of the wear-resistant layer of the Zr-ZrN-(Zr,Al,Si)N nanolayer composite coating on the values of residual stress and the nature of coating wear. The study focused on coatings with wear-resistant layer thicknesses of 2.0, 4.3, 5.9, and 8.5 μm, deposited using filtered cathodic vacuum arc deposition (FCVAD) technology. The X-ray diffraction (XRD) method based on the anisotropy of the elasticity modulus was used to find the values of the residual stress. The nature of the formation of interlayer delamination under the influence of residual compressive stress was studied using a scanning electron microscope (SEM). When the wear-resistant layers had a thickness of 2.0–5.9 μm, tensile stress formed, which decreased with an increase in the thickness of the coating. When the thickness of a wear-resistant layer was 8.5 μm, compressive stress formed. Under the action of compressive stress, periodic interlayer delamination formed, with a pitch of about 10 binary nanolayers. A mathematical model is proposed to describe the nature of the formation of interlayer delamination under the influence of compressive residual stress, including in the presence of a microdroplet embedded in the coating structure.

**Keywords:** nanolayer composite coating; filtered cathodic vacuum arc deposition (FCVAD); X-ray diffraction (XRD); residual stress; wear-resistant coating

## 1. Introduction

The internal residual stress that forms in coatings during their deposition has been a subject of research for a long time. Detailed analyses of the methods for studying stress in coatings and thin films are contained, in particular, in [1–4]. The measurement of stress in a coating with a thickness of only several micrometers, which includes various phases that differ in their properties, presents a certain challenge. The task is further complicated when it comes to coatings with nanolayer structures, including nanolayers of different phase compositions and mechanical properties. Meanwhile, residual stress has a significant effect on the functional properties of the coatings, their service life periods, and the reliability of the operation of coated products. In particular, cracks and interlayer delamination

can form in the coatings, which leads to the destruction of the structure, to chipping, and to the loss of functional properties. Compressive stress usually leads to the formation of delamination and longitudinal cracks [5,6], whereas tensile stress leads to the formation of transverse cracks [7,8]. The distribution of residual stress in the coating affects both the adhesion to the substrate and the fracture toughness [9–11].

It is clear that the formation of residual stress is associated with the process of coating deposition. From this point of view, three models of coating deposition are usually considered: formation and merging of three-dimensional (3D) islets [12], layer-by-layer deposition [13], and a combined model that considers the formation of 3D islets on a thin wetting layer [14]. Various defects on the surface of the substrate (micropores, cracks, irregularities, dirt spots, etc.) [15–21] as well as defects in the coating itself (microdroplets, interlayer delamination, etc.) have a significant influence on the coating formation process [19–21].

Depending on the type of material, two models of the formation of residual stress during the coating deposition are usually considered: Model 1, in which only tensile stress forms (in particular, the model is typical for metals with a high melting point), and Model 2, in which both tensile stress (usually on the first stage of coating deposition) and compressive stress (typical for metals with a lower melting point) form [1,4]. Another source of residual stress is the processes of recrystallization, restructuring, and diffusion and the difference in the coefficients of thermal linear expansion of various coating layers [22,23].

Various methods are used to determine the stress in the coating structure [1–4]. A number of articles have focused on the studies of residual stress in the coatings of various compositions and thicknesses. Meanwhile, the $\sin2\psi$ X-ray diffraction (XRD) method is most often used to determine the strain components for research.

Djaziri et al. [24] studied residual stress in single-layer monolithic TiN and ZrN coatings with thicknesses of 1–2 μm. The data obtained using the XRD method and other techniques (optical method and laser curvature technique) were compared, and the comparison showed a high degree of similarity of the results. Singh et al. [25] studied the residual stress on Al-SiC nanocomposite coatings, depending on the number and thickness of the layers. The multilayer structure of the coating was found to lead to an increase in the compressive residual stress compared to a monolayer structure of the coating with the same thickness. Chason et al. [26] proposed a model to predict the formation of tensile or compressive stress depending on the grain sizes, growth rate, and diffusivity. In particular, for structures with smaller grain sizes, large compressive stress was typical, whereas tensile stress formed at higher grain growth rates, and compressive stress formed at lower grain growth rates.

Bielawski et al. [27] studied TiN-Si coatings with thicknesses in a range of 1–12 μm. They found that, with an increase in the thickness, the stress transforms from compressive (for coating thicknesses less than 3 μm) to tensile (for coating thicknesses of 3–12 μm), and a higher hardness of the coating is also associated with higher values of residual stress (the results contradict the results of some other studies). Li et al. [28] studied the residual stress of a nanocrystalline $Cr_2O_3$ coating. For the given coating, the value of the residual stress does not depend on the thickness of the coating when the thicknesses exceed 0.8 μm, but at smaller thicknesses, the appropriate relation takes place. Stress decreases with a decrease in coating thickness, but in all cases, only tensile stress registered. They also argued that such coating defects as microdroplets do not significantly affect the values of residual stress. The study revealed the influence of grain sizes on the nature of stress. In particular, in [29,30], with smaller grain sizes in the coating, the compressive stress increased. Meanwhile, the influence of grain sizes on the nature of residual stress is also associated with the grain growth rate [30].

A number of studies consider residual stress evolution in coatings (in particular, biaxial stress). Veprek et al. [31] note that due to high biaxial compressive stress, the hardness of the coating can increase significantly. Specific resistance decreased with an increase in biaxial stress, while Hall mobility and carrier concentration increased. Hwang et al. [32] found that specific electric resistance decreased with an increase in biaxial compressive stress, while Hall mobility and carrier concentration increased.

In turn, residual stress is related to microstructural order in the coating structure. The results obtained by Tien et al. [33] show that residual stress and surface roughness are higher in the coatings on the samples that were located in the upper part of the chamber during the coating deposition process.

Thus, the analysis of the available studies shows that the distribution and values of residual stress depend on the chemical composition, thickness, and structure of the coating. During the formation of coatings with multilayer structures and small grain sizes, provided that the coating deposition rates (and the grain growth rate) are sufficiently low, significant compressive stress can occur with a large coating thickness. At the same time, such factors as small coating thicknesses, larger grain sizes, higher grain growth rates, and monolithic coating structures contribute to the formation of tensile stress. The purpose of the research was to study the influence of the coating thickness on the values of residual stress and to study the effect of residual stress on the nature of coating wear.

## 2. Materials and Methods

To study the relation between the values of residual stress and the thickness of the coating, four samples with the Zr-ZrN-(Zr,Al,Si)N coating were manufactured with different deposition times of the wear-resistant layers of the coating with a three-layer architecture [34–37]: 15, 30, 40, and 60 min. For the deposition of coatings, a vacuum-arc VIT-2 unit was used, which was designed for the synthesis of coatings on the substrates of various tool materials. The unit was equipped with an arc evaporator with filtration of vapor-ion flow (FCVAD) [34,35,37], which was used for the deposition of the coating on the tool to significantly reduce the formation of droplets during coating.

Prior to the deposition of the coatings, the samples were prepared as follows: initially, they were cleaned in an ultrasonic cleaner using special (environmentally friendly) detergent; then they were washed in running water, and finally in distilled water. The final cleaning of the sample surfaces was carried in the unit chamber, in a stream of gaseous plasma.

To deposit the coatings, cylindrical cathodes of the following compositions were used: Zr (99.98%) and (Al,Si) (92 at% + 8 at%; 99.96%). During the coating deposition process, 2 Zr cathodes and 1 (Al,Si) cathode were used simultaneously. The cathodes were installed in evaporators, located in one horizontal plane, and the central axes of the evaporators located at 90° to each other. The scheme of the cathode positions in VIT-2 unit and the scheme of the coating deposition process are presented in [34].

During the coating deposition, evaporators with microdroplet phase separation were used [34,35,37]. The parameters of the coating deposition are presented in Table 1.

**Table 1.** Parameters of deposition of coatings.

| Process | $p_N$ (Pa) | $U$ (V) | $I_{Al,Si}$ (A) | $I_{Zr}$(A) | $n$ (rev/min) |
|---|---|---|---|---|---|
| Pumping and heating of vacuum chamber | 0.06 | +20 | 120 | 75 | 1.1 |
| Heating and cleaning products with gaseous plasma | 2.0 | 100DC/900 AC $f$ = 10 kHz, 2:1 | 80 | - | 1.1 |
| Deposition of coating | 0.36 | −800 DC | 160 | 65 | 1.1 |
| Cooling of products | 0.06 | - | - | - | - |

Note: $I_{Zr}$ = current of zirconium cathode, $I_{Al,Si}$ = current of Al-Si cathode, $p_N$ = gas pressure in chamber, $U$ = voltage on substrate, $n$—turntable rotation speed.

For microstructural studies of samples of carbide substrates with coatings, a scanning electron microscope (SEM) FEI Quanta 600 FEG with energy-dispersive X-ray spectroscopy (EDS) was used. The studies of chemical composition were conducted using the same SEM. To perform the X-ray spectroscopy microanalysis, the study used characteristic X-ray emissions resulting from the electron bombardment of a sample.

The X-ray diffraction (XRD) analysis was performed on a PANalytical Empyrean diffractometer with CuK$\alpha$ radiation. To determine the residual stress in the coatings, the sin2$\psi$ estimation method was used [38–41]. When determining the residual stress using the sin2$\psi$ X-ray method, the lattice spacing $d_\psi^{hkl}$ for the reflection (hkl) was measured at several values of the angle $\psi$. Meanwhile, the

value $\psi = 0$ corresponds to the reflection from the planes (hkl) for grains in which the plane is parallel to the plane of the sample, and the values ($\psi \neq 0$) correspond to the reflections from the grains in which the planes (hkl) are located at an angle $\psi$ to the plane of the sample.

When the sin2$\psi$ method was used, the lattice parameters were measured at the symmetric and inclined positions of the sample with respect to the incident and reflected beams. The value of the lattice deformation differs for grains with different orientations with respect to stress. The values of the stress were calculated using the difference in the deformation. In an asymmetric survey, the informative depth decreased with an increase in the angle $\psi$. In the case under consideration, the above led to the disappearance of the diffraction maxima from the coating due to a relatively small coating thickness. Therefore, to calculate the residual stress, a technique based on the anisotropy of the elasticity modulus was used [38,39]. Since the structure of the coating is anisotropic, a more comprehensive approach to the study of residual stress is required. In the case of a nanolayer coating, it is usually an alternation of nanolayers, each of which has a columnar structure [42–46]. For this purpose, the method of unequal biaxial stress states proposed in [40,41] was used. According to the equipment manufacture, the allowance in measurements aimed to locate the centre of gravity of diffraction maximum is ±0.0001, which, when calculating the stress with the method used, corresponds to the allowance of ±3 MPa.

The hardness (HV) of coatings was determined by measuring the indentation at low loads according to the method by Oliver and Pharr [47]. This was carried out on a microindentometer microhardness tester (CSM Instruments) at a fixed load of 10 mN. For each sample, 20 microhardness measurements were carried out, the two lowest and two highest values were ignored, and then the average value was determined.

## 3. Results

### 3.1. Influence of the Coating Thickness on the Values and Nature of Residual Stress

Figure 1 shows cross sections of the coated samples under study. For Samples 1–4, the thicknesses of the wear-resistant layers are about 2.0, 4.3, 5.9, and 8.5 µm, respectively.

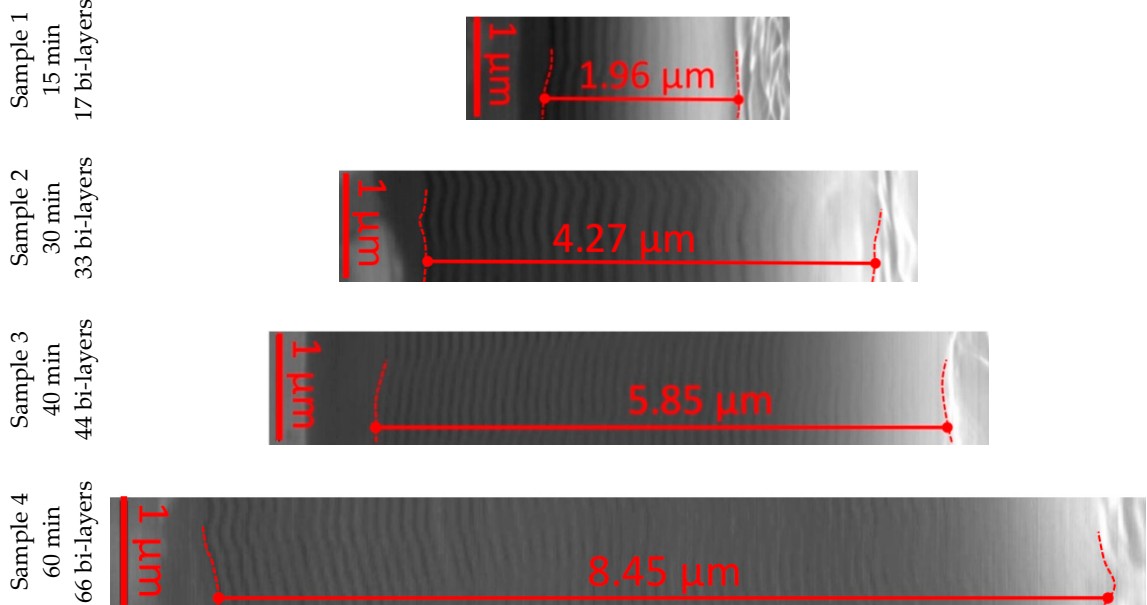

**Figure 1.** Cross sections on the Zr-ZrN-(Zr,Al,Si)N coatings under study.

Table 2 presents the microhardness values of the coated samples under study.

**Table 2.** Microhardness values of the coated samples under study.

| Sample | 1 (15 min) | 2 (30 min) | 3 (40 min) | 4 (60 min) |
|---|---|---|---|---|
| Hardness, GPa | 28.7 | 29.1 | 29.3 | 28.5 |

All the samples under study have very close, almost equal microhardness values, and that fact correlates with the data, earlier obtained for the (Ti,Al,Si)N coating [37], and proves that the deposition time (and, correspondingly, thickness) does not significantly affect the hardness of the coating.

Figure 2 presents the diffraction patterns of the coatings for Samples 1–4 under study.

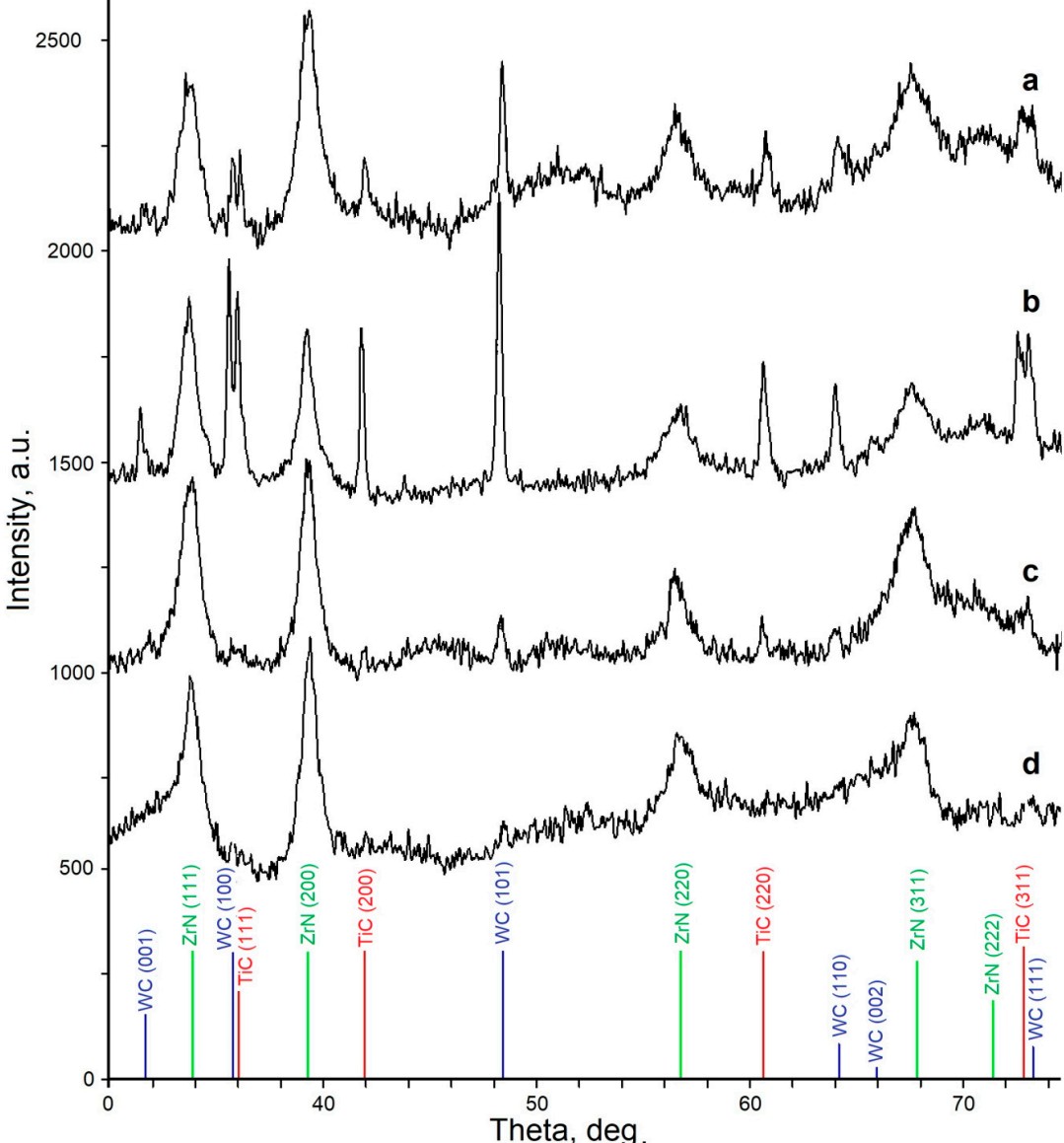

**Figure 2.** Diffraction patterns of the coatings: Sample 1 (a), Sample 2 (b), Sample 3 (c), and Sample 4 (d).

The table with the values of the lattice periods and stress calculated for the reflections characterized by elastic constants and, accordingly, different lattice deformation under the action of residual stress can be found below (Table 3).

**Table 3.** Values of the lattice periods and stress calculated for the reflections characterized by elastic constants and, accordingly, different lattice deformation.

| Sample 1 | | | | |
| --- | --- | --- | --- | --- |
| (hkl) | 2θ | | σ$_x$ (MPa) | σ$_y$ (MPa) |
| | $a_x$ | $a_y$ | | |
| (111) | 33.704 | 33.788 | 1700 | 500 |
| (200) | 39.396 | 39.292 | 1700 | 700 |
| (220) | 56.613 | 56.643 | 2000 | 1500 |
| (311) | 67.735 | 67.728 | 1800 | 650 |
| Sample 2 | | | | |
| (hkl) | 2θ | | σ$_x$ (MPa) | σ$_y$ (MPa) |
| | $a_x$ | $a_y$ | | |
| (111) | 33.896 | 33.920 | 350 | 50 |
| (200) | 39.395 | 39.375 | 1000 | 1200 |
| (220) | 56.746 | 56.680 | 2100 | 2250 |
| (311) | 67.912 | 67.843 | 350 | 250 |
| Sample 3 | | | | |
| (hkl) | 2θ | | σ$_x$ (MPa) | σ$_y$ (MPa) |
| | $a_x$ | $a_y$ | | |
| (111) | 33.736 | 33.754 | 1200 | 1200 |
| (200) | 39.351 | 39.370 | 700 | 1300 |
| (220) | 56.743 | 56.705 | 600 | −1300 |
| (311) | 67.778 | 67.572 | 1650 | 100 |
| Sample 4 | | | | |
| (hkl) | 2θ | | σ$_x$ (MPa) | σ$_y$ (MPa) |
| | $a_x$ | $a_y$ | | |
| (111) | 33.739 | 33.823 | 100 | 300 |
| (200) | 39.340 | 39.303 | 300 | −50 |
| (220) | 56.797 | 56.807 | −2300 | −1400 |
| (311) | 67.592 | 67.685 | 500 | −100 |

The data for the directions X and Y are presented in Figure 3 in a generalized graphical form.

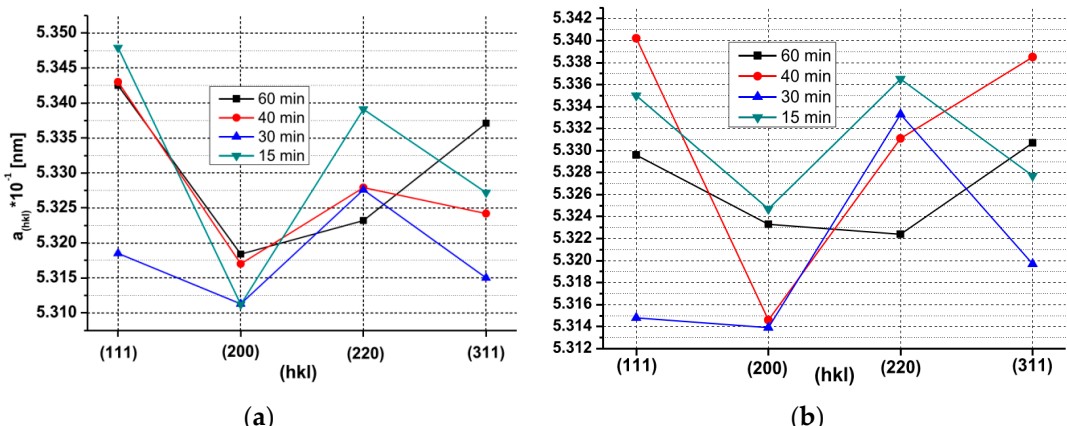

**Figure 3.** Values of the lattice periods calculated for the reflections characterized by elastic constants and, accordingly, different lattice deformation under the influence of residual stress. (**a**) Direction X, and (**b**) Direction Y.

Based on the obtained data, the calculations were run and a relation was found between the values of residual stress and the time of deposition of the Zr-ZrN-(Zr,Al,Si)N coating on a substrate of (WC + TiC) carbide (i.e., the thickness of the coating; Figure 4). The used Young's modulus values for various crystallographic directions of ZrN are presented in Table 4.

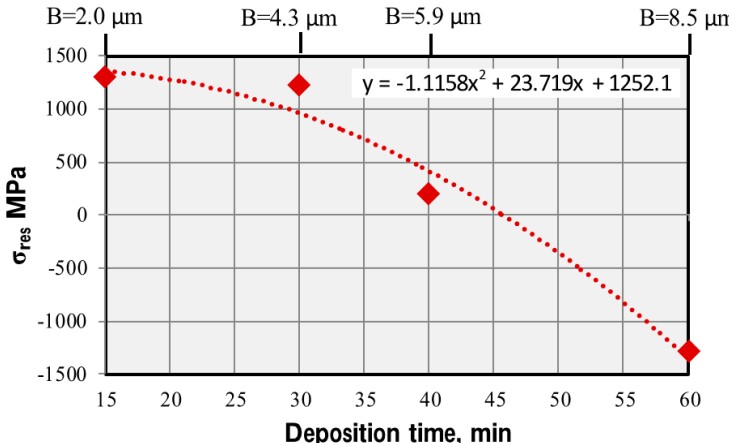

**Figure 4.** Relationship between the values of residual stress and the time of coating deposition, B – coating thickness.

**Table 4.** Used Young's modulus (E) values for various crystallographic directions of ZrN.

| hkl | 200 | 220 | 311 | 222 | 400 | 331 | 420 |
|-----|-----|-----|-----|-----|-----|-----|-----|
| E (GPa) | 418.3 | 558.7 | 446.3 | 482.4 | 418.3 | 558.7 | 437.8 |

Several studies carried out earlier (particularly [48,49]) show that tensile stress formed in the layers immediately adjacent to the substrate, whereas compressive stress in the layers is remote from the substrate. Following the analysis of the information presented in Figure 4, starting from the initial stage of the coating deposition to a certain point, high tensile stress formed, which is a natural result of the coating deposition on a substrate with a significantly lower coefficient of thermal expansion (CTE) compared to the coating itself. If for a carbide (WC + Co) substrate, CTE is $5.7–6.7 \times 10^{-6} \mathrm{K}^{-1}$ [50], then for a coating, this value reaches $8.12–14 \times 10^{-6} \mathrm{K}^{-1}$ [51,52].

The values of the stress ($\sigma_{\mathrm{tens}}$) can be defined using the following well-known relation [53]:

$$\Sigma_{\mathrm{tens}} = E(\alpha_{\mathrm{coat}} - \alpha_{\mathrm{subs}})\Delta T, \tag{1}$$

where $E$ is the Young's modulus of the coating, $\alpha_{\mathrm{coat}}$ and $\alpha_{\mathrm{subs}}$ are the coefficients of thermal linear expansion of the coating and the substrate, respectively, and $\Delta T$ is the temperature difference between the deposited coating and environment.

Another important reason for the formation of tensile stress is the beginning of merging during the initial deposition of the coating when individual islets start to collide and form grain boundaries [54,55].

Furthermore, from the substrate and layers deposited earlier, the coating layer being deposited is subjected to compressive stress caused by the mechanical action of positively charged ions bombarding the surface, possessing high kinetic energy due to the negative potential on the substrate ($\sigma_{\mathrm{comp}}$) [56,57]. The effect is similar to that of bead blasting on the surface, when compression along the normal of the surface of the substrate and the extension in its plane occur, which is inhibited by the subsurface layers of the substrate and the coating layers deposited earlier that create compressive stress in the surface layers. Another factor affecting the formation of compressive residual stress is the fact that the temperature of the nanolayer being deposited is higher than the temperature of the nanolayer deposited earlier. The deposited nanolayer begins to become cold and compresses. The values of compressive residual stress gradually increase during the deposition of each new layer until the compressive force exceeds the strength of the cohesive bond between the nanolayers and an interlayer delamination forms. In turn, that will lead to the release of compressive stress that again increases when subsequent layers are deposited. Therefore, in nanolayer coatings of large thicknesses, the formation of periodically repeating interlayer delamination can be predicted.

Thus, the total stress comprises two components:

$$\sigma_{res} = \sigma_{tens} + \sigma_{comp}. \tag{2}$$

As the coating thickness increases, the contribution of $\sigma_{tens}$ to the total stress decreases, whereas in contrast, the portion of $\sigma_{comp}$ increases, which explains the relation between the values of residual stress and the time of deposition and thickness of the coating (Figure 5).

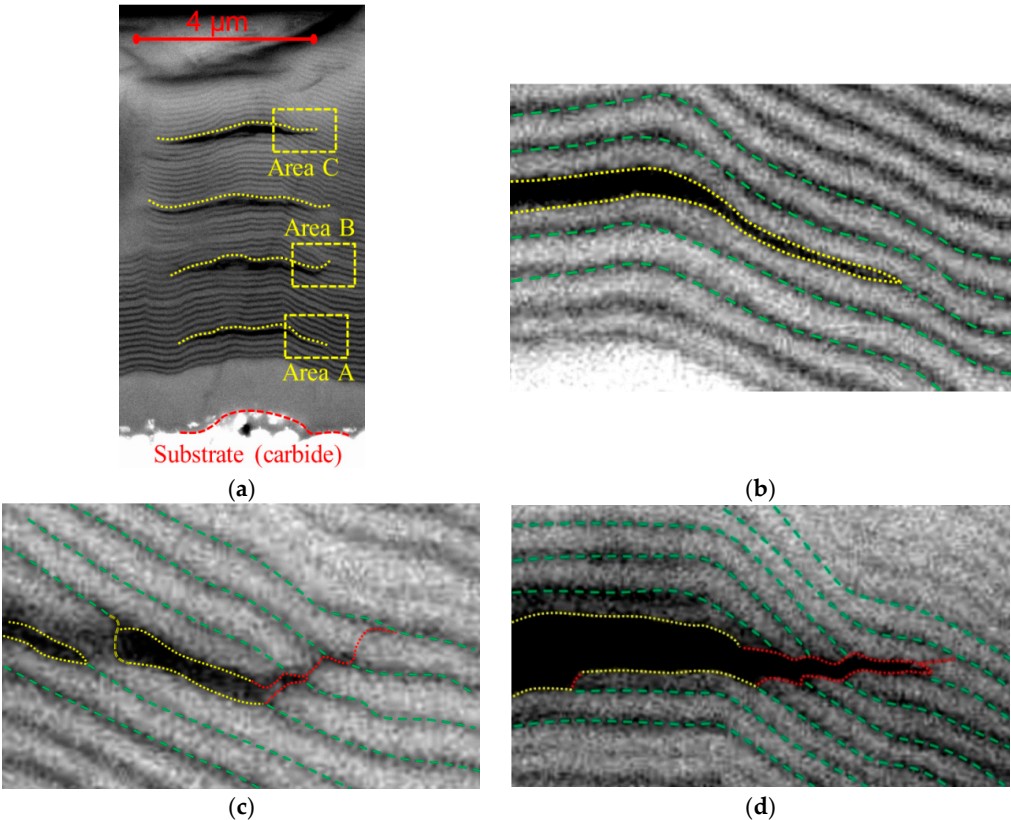

**Figure 5.** Formation of delamination between nanolayers of the coating on Sample 4. (**a**) General view of cross section, (**b**) Area A, (**c**) Area B, and (**d**) Area C.

In [13], the relation between the values of internal stress and the time of coating deposition is described as follows:

$$\frac{d(\overline{\sigma}h)}{dh} = \sigma_{xx}(h)\frac{dh}{dt} + \int_0^h \frac{\partial\sigma_{xx}(z)}{\partial t}dz, \tag{3}$$

where $h$ is the thickness of the coating at time $t$, $\sigma_{xx}(h)\frac{dh}{dt}$ determines the effect on the stress of the deposition of a new layer, and $\int_0^h \frac{\partial\sigma_{xx}(z)}{\partial t}dz$ determines the change in the stress in the layers deposited earlier. In some studies, the value is assumed to be zero (i.e., it is assumed that the stress in the layers already deposited does not change during the deposition of subsequent layers), then it is possible to obtain the following relation to determine the stress depending on the coating thickness [13]:

$$\sigma_{xx(h)} = \frac{\frac{d(\overline{\sigma}h)}{dt}}{\frac{dh}{dt}} = \frac{d(\overline{\sigma}h)}{dh}. \tag{4}$$

The studies show [1,4] that the value of $\int_0^h \frac{\partial\sigma_{xx}(z)}{\partial t}dz$ is not equal to zero (i.e., the stress in the deposited coating layers continues to change after deposition). Such a change in the stress can be

associated with various processes that continue to occur in the coating layers after their deposition: diffusion, phase transformations, grain growth, and uneven cooling of various layers.

*3.2. Influence of Compressive Residual Stress and Internal Defects in Coatings of Large Thicknesses on the Formation of Interlayer Delamination*

At a coating thickness of 8 μm, residual compressive stress of sufficiently high intensity forms in the coating structure. It is possible to find how stress affects the structure of the coating. Figure 5 shows the consideration of the formation of delamination in a coating of large thickness. Four subsequent delamination layers are shown, and there was a certain periodicity in their formation (about 10 binary nanolayers). An important point is the difference in the nature of the observed delamination, which can be associated with an increase in the value of compressive stress as the thickness of the coating increases. In particular, in Area A, located close to the substrate, only delamination was noticed, and the nanolayer structure of the coating was not violated (Figure 5a). As the coating thickness grew, the compressive stress increased, and in Area B, not only delamination but also transverse cracks through the coating nanolayers formed (Figure 5b). Finally, in Area C, located near the external surface of the coating, along with the delamination between the nanolayers, an expanding crack also formed, which destroyed the coating structure. Meanwhile, the width of the delamination itself increased and reached a thickness of 2–3 binary nanolayers (Figure 5d).

A significant effect on the nature of coating destruction under the influence of compressive stress is produced by various internal defects in the coating, in particular, microdroplets embedded in the coating structure. Figure 6a shows how delamination occurs, under the influence of a microdroplet embedded in the coating structure, transforming into a through-transverse crack. The mechanism of such a transformation was considered below. The structure of the cracks and delamination under consideration clearly reduced the strength of the coating and could lead to its early destruction during operation. In the area of the same coating, in the absence of internal defects, a sequence of delamination layers was formed without any transverse cracks (Figure 6b). In particular, in Area A, a nanolayer of the coating bends under the influence of compressive stress, and as a result, the cohesive bonds between adjacent nanolayers failed, and delamination formed.

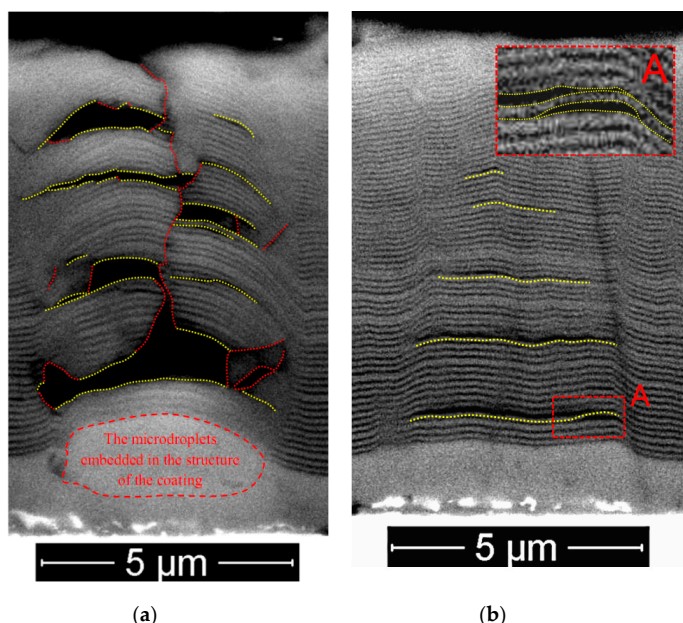

**Figure 6.** Mechanism of wear and failure. (**a**) The formation of delamination and a transverse crack under the influence of micro-drops embedded in the coating structure. (**b**) The formation of delamination only in the region without internal defects.

### 4. Discussion

Under certain conditions, the internal compressive stress leads to the formation of not only delamination and longitudinal cracks but also transverse cracks. For a better understanding of the reasons for the formation of the structure described above, the process of coating deposition was considered, in which a layer cooled down with a liquid droplet inside. Since the layer thickness was much larger than the droplet, it could be assumed that the droplet is in unbounded space. The stress–strain state is the same in any diametric section of a droplet in a vertical plane, so we considered the challenge in a cylindrical coordinate system for the environment outside of the sphere (droplet) and in a spherical coordinate system in a sphere (droplet). A droplet has an initial temperature $T^{\circ}_{sp}$. We singled out three zones (Figure 7): (1) a spherical droplet, (2) an outer spherical layer with broken lamination, and (3) a nanolayer environment (Figure 7).

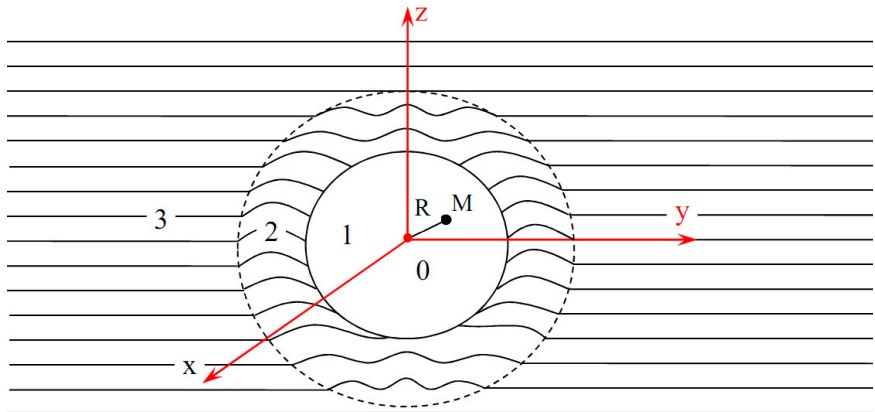

**Figure 7.** Failure of a nano-structure coating in the area surrounding an embedded droplet.

We considered the cooling of a droplet in Zone 1 under a linear time cooling regime. Temperature $T_{\kappa}$ at an arbitrary point $M$ in Zone 1 is defined by the following equation:

$$\frac{\partial T_1}{\partial t} = h_1^2 \nabla^2 T_1 = \frac{h_1^2}{r} \frac{\partial^2}{\partial r^2}(rT_1). \tag{5}$$

Through a Laplace transform (5) on time with allowance for the initial condition, we obtained the auxiliary (transformed) equation:

$$q_1^2 r \widetilde{T}_1 = \frac{\partial^2}{\partial r^2}\left(r\widetilde{T}_1\right), \ h_1^2 = \frac{k_1}{\rho_1 c_1}, \ p = h_1^2 q^2. \tag{6}$$

It is assumed that $T_c(o)$ is the temperature surrounding a droplet at the initial time point $T_1(o) = T_2(o)$, where $c_1$ is the specific heat absorption capacity for the droplet, $\rho_1$ is the density, $k_1$ is the thermal conductivity coefficient, $h_1^2$ is the temperature conductivity coefficient, and $p$ is the argument of the Laplace transform. The solution of Equation (6), limited to the droplet centre, is given by the following:

$$T_1 = r \cdot shqr. \tag{7}$$

Suppose that temperature at the outer boundary of the spherical layer (6) varies linearly with time $T_2 = Gt$, where $G$ is a constant. The boundary condition on the droplet surface is given by the following:

$$\frac{\partial T_1}{\partial r} = K(Gt - T_1). \tag{8}$$

Here, *K* depends on the ratio of thermal conductivities of Layers (1) and (2) and on the thickness of Layer (2). Suppose that a droplet radius is equal to $a = R_1$, then we have the following:

$$A = \frac{a^2 KG/p}{Kashqa + qachqa - shqa}. \tag{9}$$

The average Laplace transform for the droplet temperature by volume is calculated from the following formula:

$$<\widetilde{T_1}> = \frac{3}{a^2} \int_0^a r^2 T_1 dr = \frac{3A}{a^2} \int_0^a rshqrdr = \frac{3KG}{apq^2} \frac{qashqa - shqa}{Kashqa + qashqa - shqa}. \tag{10}$$

For small *q*, the approximate principal value for (10) has the following form:

$$<\widetilde{T_1}> \approx \frac{3KG}{apq^2} \frac{(qa)^3 \left(\frac{1}{3} + \frac{1}{30} q^2 a^2\right)}{qa\left(K_a + \frac{1}{6} Kq^2 a^2 + \frac{1}{3} q^2 a^2\right)} = G\left\{\frac{1}{p} - \left(\frac{1}{15} \frac{a^2}{h^2} + \frac{1}{3} \frac{a}{h^2 K}\right)\right\}. \tag{11}$$

The average temperature change in a droplet in time is obtained by applying the inverse Laplace transform to (11). Then, we obtained the following:

$$<T_1> = G\left\{t - \frac{a^2}{h^2}\left(\frac{1}{15} + \frac{1}{3aK}\right)\right\}. \tag{12}$$

Equation (12) shows that the temperature reduction in Zone 1 lags the temperature drop in Zone 3 because of the screening effect of Zone 2. More accurate calculations yield corrections that decrease exponentially with time. Thus, the droplet may still be in the liquid state, and the coating layers have already crystallized, so the lag in cooling leads to the formation of a concentration of residual stress in Zone 2. They can reach the limiting values with the formation of plastic zones, which are centers for crack formation.

We calculated the residual stress and deformation in Zones 1 and 2 upon cooling. The equilibrium equations in the spherical coordinate system are as follows:

$$\frac{d\sigma_{rr}^{(i)}}{dr} + \frac{2\left(\sigma_{rr}^{(i)} - \sigma_{\varphi\varphi}^{(i)}\right)}{r} = 0, \ (i = 1, 2), \tag{13}$$

where index *i* assumes the value of 1 for Zone 1 and the value of 2 for Zone 2. Here, Equation (13) is written in dimensionless form:

$$r = \frac{R}{R_1}, \ u = \frac{U}{R_1}, \ \sigma_{ij}^{(i)} = \frac{T_{ij}^{(i)}}{C_{rr}^{(i)}}, \tag{14}$$

where *R* is the dimensional radius of the considered point *N*, *U* is the shift, $T_{ij}$ is the stress, and $C_{rr}$ is the radial elasticity modulus. At the border of Zone 1, we have the following:

$$u^{(1)} = u^{(2)}, \ \gamma\sigma_{rr}^{(1)} = \sigma_{rr}^{(2)}, \ 0 \leq \gamma \leq 1. \tag{15}$$

If $\Delta T$ is the change in temperature from initial $T^0$ up to *T*, then the stress in the droplet material is as follows:

$$\sigma_{rr}^{(1)} = \frac{du^{(1)}}{dr} - k_r^{(1)} \Delta T^{(1)} + 2\beta^{(1)}\left(\frac{u^{(1)}}{r} - k_\varphi \Delta T^{(1)}\right),$$
$$\sigma_{\varphi\varphi}^{(1)} = \beta^{(1)} \frac{du^{(1)}}{dr} + \left(\lambda^{(1)} + \beta^{(1)}\right)\frac{u^{(1)}}{r} - \Delta T\left[\beta^{(1)} k_r^{(1)} + \left(\lambda^{(1)} + \beta^{(1)}\right)k_\varphi^{(1)}\right], \tag{16}$$

$$\lambda^{(1)} = \frac{c_{\varphi\varphi}^{(1)} + c_{rr}^{(1)} - c_{r\varphi}^{(1)}}{c_{rr}^{(1)}}, \quad \beta^{(1)} = \frac{c_{r\varphi}^{(1)}}{c_{rr}^{(1)}}, \tag{17}$$

where, in the general case, the coefficients of thermal conductivity in the radial direction $k_r$ and in the direction of the circle $k_\varphi$ are considered different. For an isotropic droplet, $k_r^{(1)} = k_\varphi^{(1)}$. Moreover, in the case of anisotropy, radial $C_{rr}$, circumferential $c_{\varphi\varphi}$, and shear $c_{r\varphi}$ modules are different. By substituting (16) and (17) into (13), we obtained an equation of displacements, the solution of which has the following form:

$$u^{(1)} = wr\Delta T^{(1)} + Ar^{s_1}, \quad w = \frac{-2c}{(s_2-s_1)(1-s_1)},$$
$$c = \left(1 - \beta^{(1)}\right)k_r^{(1)} - \left(\lambda - \beta^{(1)}\right)k_\theta^{(1)}, \tag{18}$$

$$\left\{ \begin{matrix} s_1 \\ s_2 \end{matrix} \right\} = \frac{-1 \pm \sqrt{1 + 8\lambda}}{2}. \tag{19}$$

Similarly, for Zone 2, we obtained the following:

$$u^{(2)} = k^{(2)}\Delta Tr + D\left[\frac{2\left(1 - 2\beta^{(2)}\right)}{1 + 2\beta^{(2)}}c_f r + \frac{1}{r^2}\right]. \tag{20}$$

$$\sigma_{rr}^{(2)} = 2\left(1 - 2\beta^{(2)}\right)D\left(c_f - \frac{1}{r^3}\right). \tag{21}$$

Random constants $A$ and $D$ are found from the coupling conditions of Zones 1 and 2 in the form of (11). With a sharp drop in temperature, it is possible that, for stress in Zone 2, the von Mises or Tresk's condition of plasticity [58,59] is met:

$$\sigma_{\varphi\varphi}^{(2)} - \sigma_{rr}^{(2)} = \lambda K, \tag{22}$$

where the limit of plasticity $K$ is as follows:

$$K^{(2)} = K_0^{(2)}\left(1 - r_3\Delta T^{(2)}\right), \tag{23}$$

where $\lambda = 1$ if $\sigma_{rr}^{(2)} < 0$ and $\lambda = -1$ if $\sigma_{rr}^{(2)} > 0$. Here, $K_0^{(2)}$ is the dimensionless coefficient of plasticity $K_0^{(2)} = {}_0^{(2)}/c_{rr}^{(2)}$ at the initial temperature, and $r_3$ characterizes the linearity of plastic softening (hardening) of the material depending on change in temperature. In plastic Zone 2, we have the following:

$$\frac{d\sigma_{rr}^{(2)}}{dr} = \frac{2\lambda K}{\xi}. \tag{24}$$

For $1 < r < \xi$, the stress is the following:

$$\sigma_{rr}^{(2)} = \frac{2}{3}\lambda K\xi^3\left(c_f - \frac{1}{r^3}\right), \tag{25}$$

where $\xi = r$ the boundary of the plastic zone. The value $c_f$ is defined from the following condition:

$$\sigma_{rr}^{(1)}\bigg|_{for\ r = c_f^{-1/3}} = 0. \tag{26}$$

The result indicates the possibility of the appearance of a plastic zone almost symmetrically around a droplet. However, due to anisotropy, plastic zones will be distributed by separate clusters, which are centers in the formation of a crack.

We considered a model for the failure of a nanolayer structure above a droplet (1, Figure 8) that has fallen on the substrate (4). The qualitative process-flow diagram assumes that a droplet, when it hits

the substrate, does not splatter but deforms and takes the form of a half ellipsoid. Further deposition of the material leads to the formation of a nanolayer arch (2) above the droplet. Near the droplet and arch, a medium (3) with longitudinal nanolayers is formed.

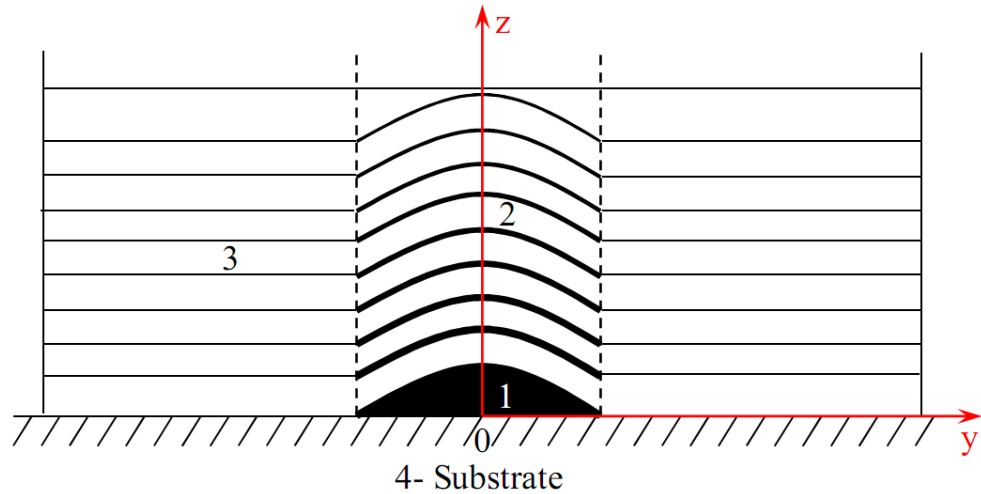

**Figure 8.** Cross section of a droplet in a nanolayer environment.

In this medium, cooling and crystallization go faster than in a droplet. Heat from the droplet rises upwards and maintains the temperature in the cylindrical volume (2) above the surrounding medium, which helps to bend the layers with convexity upwards. With further cooling, residual stress appears, applied from the side of a cooling droplet and compressing stress occurs from the side of the nanolayer coating, which contributes to the occurrence of delamination in the thickness of the nanolayers in the volume of the cylinder. The cross section of a droplet in a nanolayer medium is shown in Figure 8.

Thus, a microdroplet embedded into the structure of the coating during its deposition interacts with its structure when the coating cools down. Due to the above, compressive stress occurs, which, in turn, leads to the formation of delamination between nanolayers. If the above stress exceeds a certain value, the nanolayer arch is destroyed and a transverse crack is formed, an example of which is shown in Figure 6. The formation of such structures is typical for sufficiently thick coatings since, based on the initial conditions, the coating thickness should significantly exceed the size of an embedded droplet.

## 5. Conclusions

The article discussed the Zr-ZrN-(Zr,Al,Si)N coating with wear-resistant layer thicknesses of about 2.0, 4.3, 5.9, and 8.5 µm. The investigations of residual stress in the samples under study showed that, for coatings with wear-resistant layer thicknesses of 2.0–5.9 µm, the tensile stress was typical, which decreased as the thickness of the wear-resistant layer increased. For coatings with a wear-resistant layer thickness of 8.5 µm, sufficiently high intensity compressive stress was typical. The above result (the presence of residual compressive stress) in the latter coating is confirmed during the investigation of the formed sequences of interlayer delamination in the coating structure. The presence in the coating structure of additional stress concentrators (for example, an embedded microdroplet) in combination with a high level of compressive stress can lead to the formation of a transverse crack and destruction of the coating, which is also confirmed by the data on the appropriate mathematical model.

**Author Contributions:** Conceptualization, A.V., M.V. and A.C.; methodology, A.V., N.S. and A.A.; validation, D.L., C.S. and J.B.; formal analysis, A.C. and D.L.; investigation, N.S., A.A. and D.L.; data curation, D.L.; writing—original draft preparation, A.V.; supervision, A.V.; project administration, A.V. All authors have read and agreed to the published version of the manuscript.

**Funding:** This research was funded by Ministry of Education and Science of Russia, grant number 9.1372.2017/4.6.

**Conflicts of Interest:** The authors declare no conflict of interest.

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
