# Peer review of "Influence of the Thickness of a Nanolayer Composite Coating on Values of Residual Stress and the Nature of Coating Wear"

_coatings, doi:10.3390/coatings10010063_

Round 1

Reviewer 1 Report

Report on "Influence of the thickness of a nanolayer composite coating on values of residual stress and the nature of coating wear"

by

Alexey A. Vereschaka*, Marina A. Volosova, Anatoli Chigarev, Nikolay Sitnikov, Artem Ashmarin, Catherine Sotova, Jury Bublikov, Dmitry Lytkin

The authors report on residual stress values of different coatings with varying thickness. They further present cross-sectional images showing different failure mechanisms due to residual stresses. The major finding is a mathematical model describing the development of the residual stress state around a droplet in CAE deposited coatings. This has to my knowledge not been presented before and is of major interest.

However, there are some serious flaws within the manuscript, including missing data/information. I strongly recommend to reject the paper in its current version and would like to encourage the authors to resubmit after the following issues have been addressed:

Introduction: please include the specifics of residual stress evolution in thin films (e. biaxial stresses). Materials and Methods: there are no details on the coating deposition itself written in the manuscript such as: substrate temperature cathode configuration (which targets were used and how were they arranged within the chamber) gas composition bias voltage rotation speed applied current on the targets has there been a plasma etching process before the deposition itself? substrate pre-cleaning procedures

Please add this information

Please also specify the X-ray measurement geometries, did you use a Bragg-Brentano configuration or something different? Also, specify the used acceleration voltage and current.

Line 115: Please check the super- and subscripts. Line 130-132: The authors write about hardness testing, however, there are no hardness values reported in the manuscript. Besides this, I would assume that for a fixed load of 100 mN the indentation depth exceeds 1/10th of the film thickness, please consider to lower the maximum value. Figure 2: The indexing of ZrN is very hard to read/blurry. Please provide a high-resolution image of this figure. Furthermore, the x-axis seems to start with 2θ = 30°. Please also provide at least one representative sin2ψ plot to Figure 3: the labeling of the y-axis seems to be incorrect There is no information on the sequence of the individual layers as well as on the single-layer thickness. Please consider to at least calculate the bi(tri?)-layer thickness by dividing the total thickness of the coating by the number of total rotations during the deposition process and comment on the sequence of the layers. Figure 4: are these values mean values from different 2θ positions? There is no information given on Young’s modulus values used to calculate the residual stress. Please, also give the reader an idea of the thermal expansion coefficients of the coating and substrate (line 163) and the resulting stress due to cooling. Line 176-178: Is the substrate heated? Why should there be a temperature difference between those layers? Please explain this in detail Line 197: Phrases like ...and so on should be avoided as they leave an impression of lack of interest. Please name “and so on”. Calculation of residual stresses in the vicinity of a microdroplet:

I like the idea of this mathematical model, however, it is, of course, harder to follow these formulae than an illustration showing what is going on during the cooling of such a droplet.

Therefore I would suggest, in order to enhance the readability of the manuscript, to make such an illustration showing the temperature profile and the resulting stress profiles (cross-sectional as overlay).

Author Response

Reviewer 1

The authors report on residual stress values of different coatings with varying thickness. They further present cross-sectional images showing different failure mechanisms due to residual stresses. The major finding is a mathematical model describing the development of the residual stress state around a droplet in CAE deposited coatings. This has to my knowledge not been presented before and is of major interest.

However, there are some serious flaws within the manuscript, including missing data/information. I strongly recommend to reject the paper in its current version and would like to encourage the authors to resubmit after the following issues have been addressed:

A_ The authors apologize for the errors and inaccuracies and are grateful to the Reviewer for a number of valuable comments. The authors agree with the Reviewer that the article required serious additions and changes, without which it was not worthy of publication. The authors hope that the changes and additions made will help to improve the quality of the article.

R_ Introduction: please include the specifics of residual stress evolution in thin films (e. biaxial stresses).

A_ Information added.

R_ Materials and Methods: there are no details on the coating deposition itself written in the manuscript such as: substrate temperature cathode configuration (which targets were used and how were they arranged within the chamber) gas composition bias voltage rotation speed applied current on the targets has there been a plasma etching process before the deposition itself? substrate pre-cleaning procedures. Please add this information

A_ Information added.

R_ Please also specify the X-ray measurement geometries, did you use a Bragg-Brentano configuration or something different? Also, specify the used acceleration voltage and current.

We used the Bragg-Brentano geometry, a glow current of 40 mA, and an accelerating voltage of 40 kV.

R_ Line 115: Please check the super- and subscripts.

A_ Corrected

R_ Line 130-132: The authors write about hardness testing, however, there are no hardness values reported in the manuscript. Besides this, I would assume that for a fixed load of 100 mN the indentation depth exceeds 1/10th of the film thickness, please consider to lower the maximum value.

A_ The authors apologize for the inaccuracy. The measurement load was not 100, but only 10 mN. Since with such a small load, the indenter imprint is sometimes not very clear, which leads to a large dispersion of values, the following method was used: For each sample, 20 microhardness measurements were performed, the two smallest and two largest values were discarded, after which the average value was determined.  In fact, microhardness measurements were not initially performed. However, now they are held. Hardness measurement results added.

R_ Figure 2: The indexing of ZrN is very hard to read/blurry. Please provide a high-resolution image of this figure. Furthermore, the x-axis seems to start with 2θ = 30°. Please also provide at least one representative sin2ψ plot to

A_ Stresses were measured not by the sin2Ψ method, but by a method based on the anisotropy of the elastic modulus; therefore, there are no sections with measuring the location of the diffraction maximum at Ψ ≠ 0.

R_ Figure 3: the labeling of the y-axis seems to be incorrect

A_ Corrected

R_ There is no information on the sequence of the individual layers as well as on the single-layer thickness. Please consider to at least calculate the bi(tri?)-layer thickness by dividing the total thickness of the coating by the number of total rotations during the deposition process and comment on the sequence of the layers.

A_ Information added (in Figure 1)

R_ Figure 4: are these values mean values from different 2θ positions? There is no information given on Young’s modulus values used to calculate the residual stress. Please, also give the reader an idea of the thermal expansion coefficients of the coating and substrate (line 163) and the resulting stress due to cooling.

A_ Information added

R_ Line 176-178: Is the substrate heated? Why should there be a temperature difference between those layers? Please explain this in detail

A_ With the technology used in this work, there is no permanent heating of the substrate during the deposition process. The freshly deposited layer begins to cool due to heat radiation and the transfer of thermal energy to the gas medium and to the lower layers. Accordingly, the newly deposited layer will have a slightly higher temperature than the previously deposited layer. To maintain the temperature of the substrate and ensure sufficient adhesion, the substrate is periodically heated by a stream of gas plasma. However, a decrease in temperature occurs between these heaters.

R_ Line 197: Phrases like ...and so on should be avoided as they leave an impression of lack of interest. Please name “and so on”.

A_ “And so on” has been removed from the text. Using “and so on”, the authors had in mind that theoretically there may be some other factors (besides those listed) that influence the change in stresses. Having examined various aspects of the coating deposition process, the authors added another factor - "uneven cooling of various layers".

R_ Calculation of residual stresses in the vicinity of a microdroplet:

I like the idea of this mathematical model, however, it is, of course, harder to follow these formulae than an illustration showing what is going on during the cooling of such a droplet.

Therefore I would suggest, in order to enhance the readability of the manuscript, to make such an illustration showing the temperature profile and the resulting stress profiles (cross-sectional as overlay).

A_ In this article, the team of authors, consisting of scientists in the field of coating deposition, materials scientists and mathematicians, begins a series of studies devoted to studying the influence of various factors (coating thickness, nanolayer thickness, depending on the rotation speed of the turntable, chemical composition of the coating, deposition process parameters, and t .d.) by the magnitude and distribution of residual stresses. As a “global” problem, we would like to get a mathematical model that describes the influence of various factors of the deposition process on various properties of the coating. Understanding the complexity of this task (given, in particular, the fact that the coating deposition process is stochastic, the anisotropy of the coating and the difference in coating properties at different points of the coated surface are obvious), the authors nevertheless hope to take steps to solve it. Thus, the authors honestly acknowledge that the model presented in this article is conceptual in nature, allowing a qualitative assessment of the process. At the next stages, we plan to proceed to the forecasting of quantitative indicators (for example, the value of residual stresses). Such (calculated) values could be compared with the measured ones and thus verify the adequacy of the model.

Reviewer 2 Report

The article presented by A. Vereschaka et al, shows an interesting study on the residual stress influence on wear. The study is well conducted and shows quite insightful results on the general mechanisms. Some minor remarks are given as follow:

Can the authors provide Sin2(chi) measurements, for the peaks of ZrN(311) and other high angle peaks in the samples?. This is in order to corroborate the allocated stress assumptions. Delamination of interfaces is clearly associated to the allocated stress, as the authors show. However, the general roughness of the substrate vs thickness of the first few layers, in which case, the miss-match/crack propagation is associated with these interfaces. Can the authors discuss this in detail?

Author Response

Reviewer 2

The article presented by A. Vereschaka et al, shows an interesting study on the residual stress influence on wear. The study is well conducted and shows quite insightful results on the general mechanisms. Some minor remarks are given as follow:

A_ The authors thank the Reviewer for the time spent studying their work and valuable recommendations.

R_ Can the authors provide Sin2(chi) measurements, for the peaks of ZrN(311) and other high angle peaks in the samples?.

A_ Unfortunately, at the moment there is no opportunity to conduct such studies, but the authors will definitely take advantage of this reviewer's recommendation in their future work.

R_ This is in order to corroborate the allocated stress assumptions. Delamination of interfaces is clearly associated to the allocated stress, as the authors show. However, the general roughness of the substrate vs thickness of the first few layers, in which case, the miss-match/crack propagation is associated with these interfaces. Can the authors discuss this in detail?

A_ Since the authors consider coatings for metal cutting tools, carbide cutting inserts are used as a substrate, and there is no additional surface preparation (other than that provided for by the cutting tool manufacturing technologies). As defects affecting the formation of zones with an increased level of residual stresses, one can consider both initial surface defects (irregularities, protruding carbide grains, microcracks, etc.), defects that formed during the deposition process (primarily microdroplets, and - residual microparticles of pollution). Previous studies show that, since the plasma flow is not completely homogeneous, and the deposition process is stochastic, the thickness of the deposited nanolayer is unstable and may be zero in some areas, that is, there are “holes” in each nanolayer. The deposited Zr-ZrN- (Zr, Al, Si) N coating implies the presence of a first (adhesive) layer consisting of pure zirconium. The reason for this sufficiently soft and at the same time plastic layer is not only providing good adhesion of the coating to the substrate, but also "healing" of various defects of the substrate, that is, there is some leveling of the surface. The specified adhesive layer has a thickness of 20-50 nm and, accordingly, may also contain some leaks. Then, a harder and less plastic ZrN transition layer (300-600 nm thick) is deposited, after which a wear-resistant layer with maximum hardness is deposited directly. Thus, the effect of the initial defects and substrate roughness on the internal residual stresses in the wear-resistant layer is somewhat reduced. Moreover, the microdroplets formed during the deposition process have a greater effect (as shown in Figure 6).

Reviewer 3 Report

The authors should in more detail discuss the effect of substrate temperature increase or change, which can affect residual stress decrease. In Fig. 4 standard deviations should be presented to reveal significance of the differences obtained. The same is for Fig. 3. Actually the points in the graphs should not be connected, better approximated with a function. Can somehow the atomic ratios between the elements in the coatings Zr-ZrN-(Zr,Al,Si)N be specified in the manuscript The authors should discuss in more details why the considered relatively narrow thickness range, what happens if the thickness is less or more thick.

Author Response

Reviewer 3

A_ The authors thank the Reviewer for the time spent studying their work and valuable recommendations.

R_ The authors should in more detail discuss the effect of substrate temperature increase or change, which can affect residual stress decrease.

A_ The authors completely agree with the Reviewer that the temperature of the substrate during deposition of the coating is of great importance on the properties of the coating (including the nature and magnitude of the residual stresses). In this case, the measurement of the temperature of the substrate itself is a very complex and not completely solved task (due to the rapid movement of the tool with samples, a significant difference in temperature in different areas of the tool, possible contamination of the glasses and a large number of other factors). The authors plan to consider the influence of various factors of the deposition process (in particular, the rotation speed of the turntable, substrate temperature and gas pressure) on the value of the residual stresses in one of the subsequent works.

R_ In Fig. 4 standard deviations should be presented to reveal significance of the differences obtained.  The same is for Fig. 3.

A_ Since the measurement deviations is only ± 3 MPa (based on the data of the equipment manufacturer), the authors considered it inappropriate to supplement the graphs with “deviations bars”, since this would be impossible to see. At the same time, the corresponding information on measurement deviations was added to the Materials and Methods section.

R_ Actually the points in the graphs should not be connected, better approximated with a function.

A_ Figure 4 has been modified in accordance with the recommendations of the Reviewer.

R_ Can somehow the atomic ratios between the elements in the coatings Zr-ZrN-(Zr,Al,Si)N be specified in the manuscript

A_ Unfortunately, in this study, we did not measure the ratio of elements in the coating. However, the previously performed measurements for the coating deposited under similar conditions are (excluding the nitrogen content, only the ratio of metals to silicon): 70 at% Zr + 25 at% Al + 5at% Si

R_ The authors should discuss in more details why the considered relatively narrow thickness range, what happens if the thickness is less or more thick.

A_ Since the considered range of coating thicknesses (2.0 - 8.5 μm) generally coincides with the range of applied coating thicknesses (less than 2 μm, the coating does not show a noticeable increase in the tool durability period, and with thicknesses of 10 μm or more, the coating breaks rather quickly, possibly including under the influence of a high level of residual stresses), the authors considered it appropriate to limit themselves to the considered range of thicknesses. Earlier in our work (in particular, doi: 10.3390 / coatings9110730), we considered coatings with thicknesses up to 16 μm. Unfortunately, then stress measurements were not carried out. However, studying the nature of the destruction of thick coatings suggests the presence of high intensity compressive residual stresses in the structure. In the future, the authors have plans for a more detailed study of coatings of large thickness (up to 20 μm) with the mandatory measurement of residual stresses, and also to explore ways to reduce the level of residual stresses in such coatings.

Reviewer 4 Report

Title of article is accord with the body of the text. Article is presenting the Influence of the thickness of a nanolayer composite coating on values of residual stress and the nature of coating wear.

Introduction provides sufficient background and includes all relevant references. The research problem is well clarified through the text. The scientific problem analysis is well elaborated. Research design is appropriate. The methods used to gain results are adequately described. The main propositions of the research are reasoning. Results clearly presented. Conclusions are supported by the results. The research method is well elaborating and comprehensive.

The text is written in a good and comprehensive English.

Author Response

Reviewer 4

R_ Title of article is accord with the body of the text. Article is presenting the Influence of the thickness of a nanolayer composite coating on values of residual stress and the nature of coating wear.

Introduction provides sufficient background and includes all relevant references. The research problem is well clarified through the text. The scientific problem analysis is well elaborated. Research design is appropriate. The methods used to gain results are adequately described. The main propositions of the research are reasoning. Results clearly presented. Conclusions are supported by the results. The research method is well elaborating and comprehensive.

The text is written in a good and comprehensive English.

A_ The authors are grateful to the Reviewer for the high appreciation of their work

Round 2

Reviewer 1 Report

I would like to thank the authors for adding information to the manuscript.

Reviewer 2 Report

The article has been amended according to my suggestions. 

Reviewer 3 Report

from my point of view the manuscript can be accepted.